# Mouse-Geneformer: A deep learning model for mouse single-cell transcriptome and its cross-species utility

Keita Ito[1], Tsubasa Hirakawa[2], Shuji Shigenobu[3,4]*, Hironobu Fujiyoshi[5]*, Takayoshi Yamashita[6]*

1 Graduate School of Engineering, Chubu University, Kasugai, Aichi, Japan, 2 Department of Artificial Intelligence and Robotics, Center for Mathematical Science and Artificial Intelligence, Chubu University, Kasugai, Aichi, Japan, 3 Trans-Scale Biology Center, National Institute for Basic Biology, Okazaki, Aichi, Japan, 4 Life Science Center for Survival Dynamics, Tsukuba Advanced Research Alliance (TARA), University of Tsukuba, Tsukuba, Ibaraki, Japan, 5 Department of Computer Science, Chubu University, Kasugai, Aichi, Japan, 6 Department of Artificial Intelligence and Robotics, Chubu University, Kasugai, Aichi, Japan

* takayoshi@isc.chubu.ac.jp (TY), fujiyoshi@isc.chubu.ac.jp (HF), shige@nibb.ac.jp (SS)

## Abstract

Deep learning techniques are increasingly utilized to analyze large-scale single-cell RNA sequencing (scRNA-seq) data, offering valuable insights from complex transcriptome datasets. Geneformer, a pre-trained model using a Transformer Encoder architecture and human scRNA-seq datasets, has demonstrated remarkable success in human transcriptome analysis. However, given the prominence of the mouse, *Mus musculus*, as a primary mammalian model in biological and medical research, there is an acute need for a mouse-specific version of Geneformer. In this study, we developed a mouse-specific Geneformer (mouse-Geneformer) by constructing a large transcriptome dataset consisting of 21 million mouse scRNA-seq profiles and pre-training Geneformer on this dataset. The mouse-Geneformer effectively models the mouse transcriptome and, upon fine-tuning for downstream tasks, enhances the accuracy of cell type classification. In silico perturbation experiments using mouse-Geneformer successfully identified disease-causing genes that have been validated in in vivo experiments. These results demonstrate the feasibility of analyzing mouse data with mouse-Geneformer and highlight the robustness of the Geneformer architecture, applicable to any species with large-scale transcriptome data available. Furthermore, we found that mouse-Geneformer can analyze human transcriptome data in a cross-species manner. After the ortholog-based gene name conversion, the analysis of human scRNA-seq data using mouse-Geneformer, followed by fine-tuning with human data, achieved cell type classification accuracy comparable to that obtained using the original human Geneformer. In in silico simulation experiments using human disease models, we obtained results similar to human-Geneformer for the myocardial infarction model but only partially consistent results for the COVID-19 model, a trait unique to humans (laboratory mice are not susceptible to SARS-CoV-2). These findings suggest the potential for cross-species application of the Geneformer model while emphasizing the importance of species-specific models for capturing the full complexity of disease

**Data availability statement:** The code of mouse-Geneformer is available on GitHub at https://github.com/machine-perception-robot-ics-group/Mouse-Geneformer. The mouse-Genecorpus-20M is also publicly available on Hugging Face at https://huggingface.co/datasets/MPRG/Mouse-Genecorpus-20M.

**Funding:** This work was supported in part by NIBB Collaborative Research Program (24NIBB462 to H.F.). The funders had no role in study design, data collection and analysis, decision to publish, or preparation of the manuscript.

**Competing interests:** The authors have declared that no competing interests exist

mechanisms. Despite the existence of the original Geneformer tailored for humans, human research could benefit from mouse-Geneformer due to its inclusion of samples that are ethically or technically inaccessible for humans, such as embryonic tissues and certain disease models. Additionally, this cross-species approach indicates potential use for non-model organisms, where obtaining large-scale single-cell transcriptome data is challenging.

## Author summary

Researchers have developed Geneformer, a powerful tool that utilizes advanced deep learning techniques and large-scale single-cell transcriptome data to analyze human cell genetic activity. However, given the extensive use of mice (*Mus musculus*) in medical and biology research, there is a need for a similar tool tailored to this model organism. To address this gap, we developed mouse-Geneformer, an adaptation of Geneformer trained on a large dataset of mouse single-cell RNA sequencing data obtained from 20 million cells. Mouse-Geneformer demonstrates high accuracy in identifying distinct cell types and predicting disease-causing genes in gene manipulation simulation experiments. Moreover, mouse-Geneformer exhibited comparable accuracy to the original human Geneformer, even when applied to human cell data, suggesting its potential for cross-species use. For instance, it performed well in studying heart disease but was less consistent with COVID-19, likely due to the differences between species in how they react to the virus. Overall, mouse-Geneformer could be a valuable resource for studying not only mice but also other animals, especially when large-scale data are challenging to obtain. Furthermore, this cross-species approach may probe beneficial in human research, especially for tissues that are difficult to access, such as embryonic samples.

## Introduction

Single-cell RNA sequencing (scRNA-seq) is a powerful technique that quantifies gene expression profiles at the individual cell level [1]. Recent technological advances in scRNA-seq have facilitated the rapid expansion of transcriptomic data and enabled the simultaneous analysis of thousands of single cells. This capability has significantly enhanced our understanding of developmental processes and disease mechanisms by revealing previously hidden heterogeneous cellular populations and novel cell types. scRNA-seq is being applied to a wide variety of experiments across different organisms, resulting in a rapid growth of scRNA-seq databases. These extensive datasets offer tremendous potential for collective use, providing valuable insights into genetic architecture and furthering our knowledge of cellular and molecular biology.

Deep learning, a recent advancement in artificial intelligence, has been successfully applied to numerous problems involving large datasets, and has emerged as a promising tool for analyzing scRNA-seq data [2–9]. This approach is particularly effective at extracting valuable information from noisy, heterogeneous, and high-dimensional transcriptome data. Among the successful methods in this domain is Geneformer, a pre-trained model that employs a Transformer Encoder architecture [10], similar to BERT [11], a widely used attention-based deep learning model in natural language processing. Geneformer utilizes the attention mechanism to calculate the relationships between genes and the context within "cell sentences", enabling it to comprehend context-dependent genetic network dynamics. By fine-tuning

Geneformer on specific downstream tasks using limited data, researchers can achieve accurate cell type classification. Additionally, Geneformer facilitates in silico simulations of gene manipulation experiments, thereby streamlining the identification of disease-causing genes and advancing our understanding of genetic networks and disease mechanisms.

The mouse, *Mus musculus*, is the foremost mammalian model for studying human biology and disease [12]. Extensive knowledge has been accumulated about mouse physiology, anatomy, and genetics. Methods for genetic manipulation, such as creating transgenic, knockout, and knockin animals, which are ethically and technically challenging in humans, have significantly advanced mouse research. These tools have led to a dramatic increase in the use of mice as model organisms. scRNA-seq experiments have also been actively applied to mouse studies, resulting in a rapid accumulation of scRNA-seq data. In this context, a deep learning model of the mouse transcriptome would greatly benefit mouse studies, and Geneformer presents a promising candidate for this purpose. However, the original Geneformer is modeled on the human transcriptome. Predicting disease-causing genes in mice using a human-based Geneformer is not straightforward. Therefore, a mouse version of Geneformer (mouse-Geneformer) is in high demand.

This paper aims to construct a mouse version of Geneformer, a deep learning model trained on mouse scRNA-seq data. We then evaluate the usefulness of mouse-Geneformer for various downstream analyses such as cell type classification and in silico perturbation experiments. We also explore the potential for cross-species application of the mouse-Geneformer. If successful, mouse-Geneformer could be used for human studies, where some samples are inaccessible due to ethical and technical constraints, and for non-model organisms, where large-scale scRNA-seq data are not available.

## Materials and methods

### Construction of the mouse-Geneformer

**Architecture.** We constructed mouse-Geneformer, a deep learning model aimed at predicting the gene network of normal mice. We developed the mouse-Geneformer following the original human version of Geneformer, a context-aware, attention-based deep learning model, which was pretrained on large-scale transcriptome data comprising approximately 30 million human single-cell RNA-seq data, developed by Theodoris et al [10]. The architecture and transfer learning strategy employed in our development of mouse-Geneformer are the same as those of the original human Geneformer, with minor modifications. An overview diagram of the mouse-Geneformer is shown in Fig 1. Instead of human RNA-seq data, we used mouse single-cell RNA-seq data as an input corpus. The transcriptome of each single cell was presented to the model by using the Rank Value Encoding method that was also developed in the original human Geneformer [10]. Finally, a Geneformer model constructed with Transformer Encoder using the created cell texts is pre-trained with self-supervised learning to build the mouse-Geneformer.

**Construction of mouse-Genecoupus-20M.** We constructed a large-scale dataset of mouse single-cell RNA-seq data, termed mouse-Genecorpus-20M, derived from healthy wild-type *Mus musculus* strains to learn the gene network of healthy wild-type mouse. The dataset was compiled from publicly available mouse single-cell RNA-seq data sourced from multiple databases: PanglaoDB [13], Single Cell Expression Atlas [14], Single Cell Portal [15], ENCODE project [16], 10x Genomics [17], and CELLxGENE [18]. We used datasets generated by droplet-based sequencing platforms such as Chromium (10x Genomics), Drop-seq, DropNuc-seq, MIRACL-seq, and inDrops. In total, 1,089 RNA-seq datasets were compiled. The complete list of datasets, along with their metadata, is provided in S1

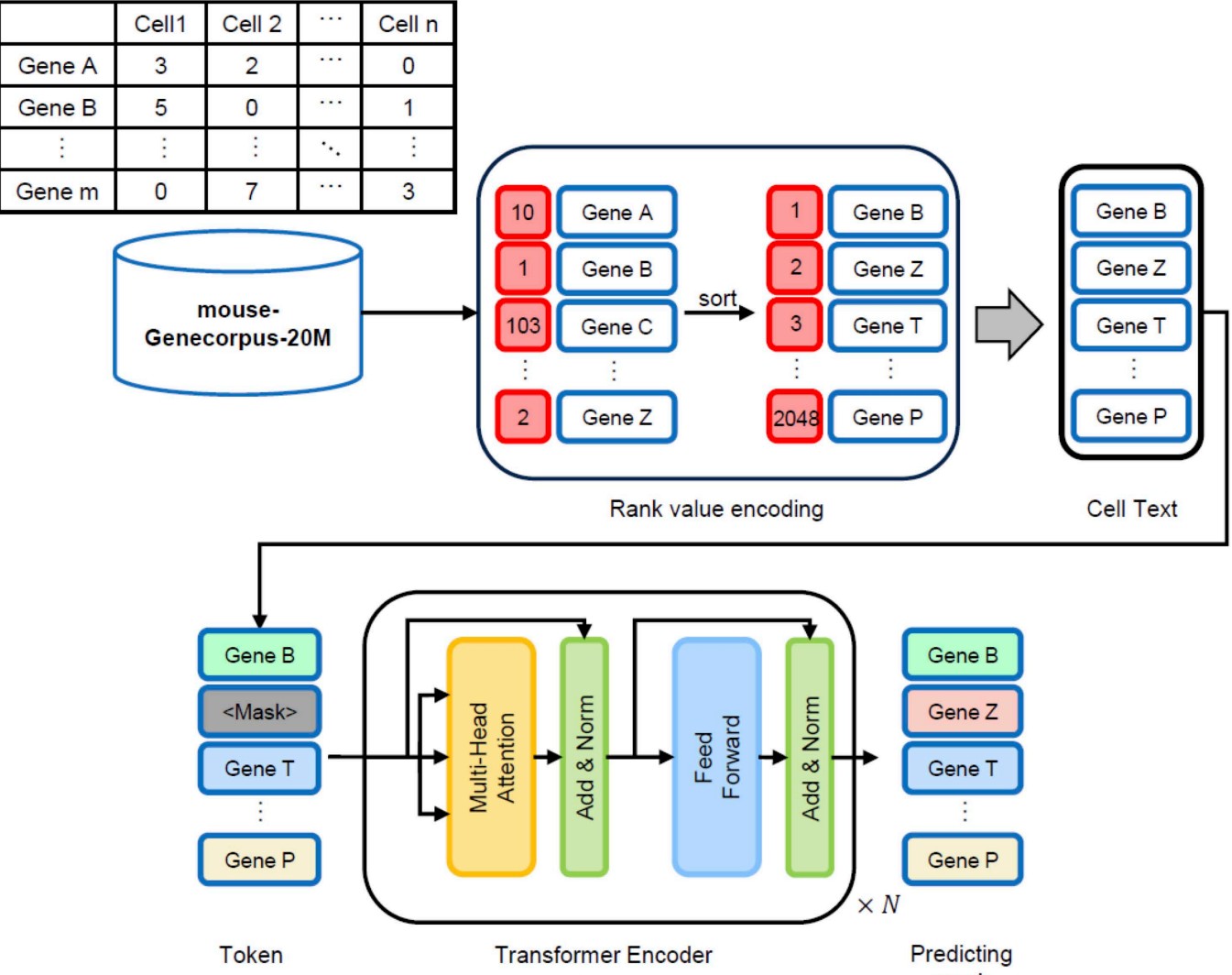

**Fig 1. Overview of the mouse-Geneformer.** mouse-Genecorpus-20M is a large mouse transcriptome dataset obtained from single-cell RNA-seq experiments, Rank value encoding is a method for generating "cellular sentences", Transformer Encoder is the architecture of the mouse-Geneformer.

Table, and an overview is shown in Fig 2. These datasets collectively represent single-cell transcriptome data from 119 million cells before filtering (see below). The majority of the datasets were derived from C57BL/6 mice or closely related strains. However, the precise genetic composition of the dataset remains uncertain due to the limited availability of strain annotations in public databases. Given the substantial genetic and transcriptional differences among mouse strains, users should carefully consider these factors when applying this model, particularly in contexts where genetic diversity may influence the results.

These single-cell datasets were stored and processed in HDF5 format loom files, H5AD files, and pickle files using the tools and libraries including scanpy, loompy, pandas, numpy, pickle, and scipy. The raw gene expression data is structured in matrices where rows represent genes, columns represent cells, and the gene expression levels are stored in the cells of the matrix. The dataset uses Ensembl IDs for gene names. Since the Ensembl ID version varied across datasets, the version information was ignored.

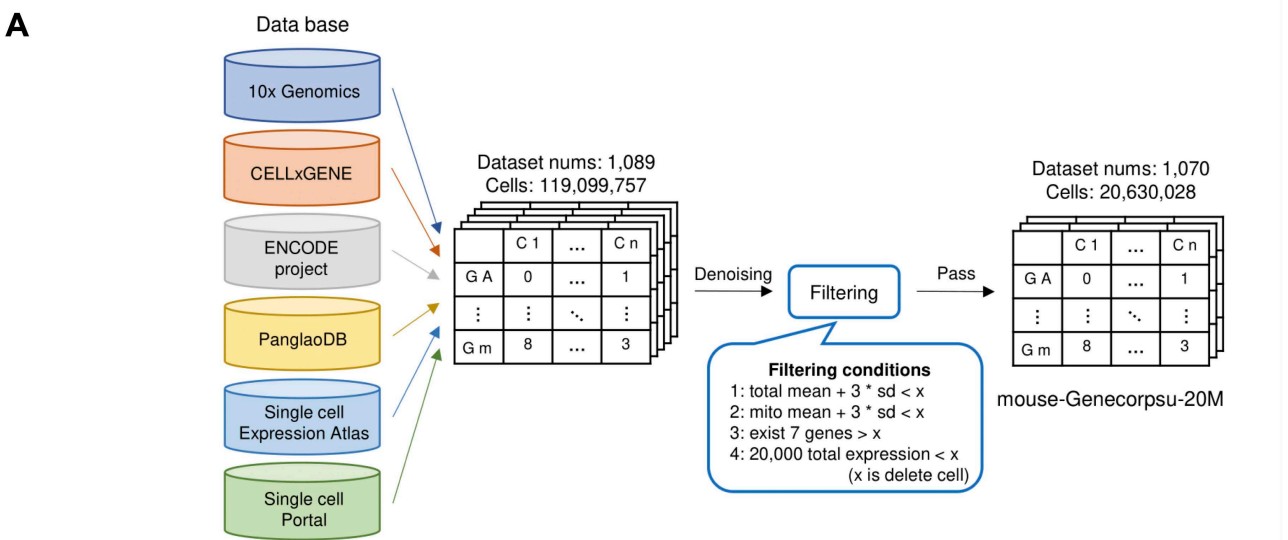

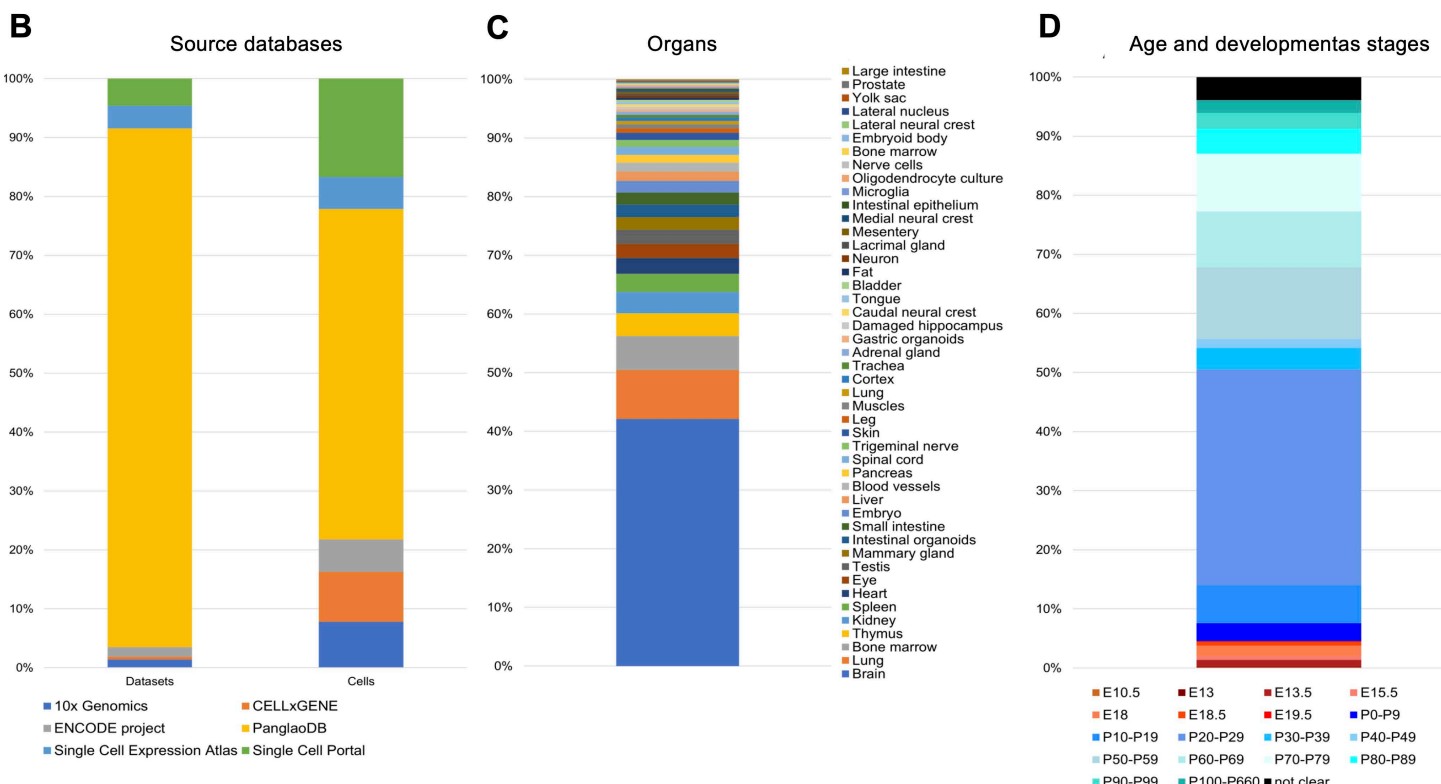

**Fig 2. Building mouse-Genecourpus-20M from single-cell RNA-seq datasets.** (A) Overview of the process for creating mouse-Genecorpus-20M. (B) Sources of the databases used for building the mouse-Genecorpus-20M. The vertical axis shows the percentage of the number of datasets or cells derived from each database. (C) Distribution of mouse organs included in the mouse-Genecorpus-20M. The vertical axis indicates the percentage of cell numbers corresponding to each organ. (D) Age and developmental stages of mice used in the mouse-Genecorpus-20M. The vertical axis indicates the percentage of cell numbers corresponding to mouse age or embryonic stage. Labels starting with "E" represents embryonic and "P" represents postnatal stages.

Starting with 119 million cells from the compiled 1,089 datasets, we applied filtering based on metadata and quality. Since mouse-Genecorpus-20M is intended for learning the gene network of normal mice to build a reference genetic architecture, we excluded single-cell data with high mutation loads that could cause restructuring of gene networks, such as those from cancer cells or immortalized cell lines. Quality filtering also addressed artifacts common in droplet-based single-cell data, such as ambient RNA [19,20], doublets [21,22], and data derived from empty droplets. The specific filtering conditions applied were as follows: 1) For each dataset, cells with total gene expression levels more than three standard deviations from the mean were excluded. 2) Cells with mitochondrial RNA expression levels more than three standard deviations from the mean were excluded. 3) Cells with fewer than seven detected genes per cell were removed. 4) Cells where the sum of the expression levels of all genes per cell exceeded 20,000 were excluded, as excessively high expression levels may indicate artifacts such as doublets or multiplets. After applying these filters, the final dataset was 1,070 RNA-seq datasets and encompassed 20,630,028 cells. The mouse-Genecorpus-20M dataset is formatted in Apache Arrow and is available for download from the repository (https://huggingface.co/datasets/MPRG/Mouse-Genecorpus-20M). It can be accessed using the Python library "datasets" for downstream applications.

**Pretraining of the mouse-Geneformer.** Pretraining was conducted following the procedures described previously for the original human Geneformer [10], with some modifications. Details are described in the Results section. Briefly, the mouse-Genecorpus-20M dataset was processed using Rank Value Encoding to extract genes that capture cell features, forming cell sequences with these genes as tokens. A special token called [CLS] is added for classifying cell sequences. The resulting data, consisting of cell sequences with token IDs, is used as input for pre-training the mouse-Geneformer. The pretraining employed a masked token prediction task, where 15% of the tokens in the input sequences were randomly masked, and the model was trained to predict these masked tokens using the remaining tokens. This task optimizes the network, allowing the mouse-Geneformer to learn the relationships and expression patterns of genes in mice.

We conducted the pretraining of the mouse-Geneformer using mouse-Genecorpus-20M under the conditions shown in Table 1. The training utilized 8 NVIDIA V100 GPUs, each with 32 GB of memory, and the process took approximately 2 days

## Fine-turning of the mouse-Geneformer

Fine-tuning the mouse-Geneformer enables accurate classification of cell types and disease types, even with limited mouse single-cell data, such as data containing rare cells, disease-specific data, or organ-specific data. This process leverages the knowledge gained from pretraining on mouse-Genecorpus-20M to predict gene networks accurately. Fine-tuning involves adding a classification layer to classify the [CLS] token in the final layer of the mouse-Geneformer. The model is initialized with the weights from the pre-trained mouse-Geneformer and further trained using a small amount of specific data. When the fine-tuning task is similar to the pretraining task, some layers of the Transformer Encoder Block are fixed to enhance generalization performance. To mitigate overfitting, which can occur due to the limited amount of data, the number of training iterations is reduced.

## Cell type classification using the mouse-Geneformer and conventional methods

To evaluate mouse-Geneformer, we compared its performance on cell type classification tasks with that of conventional methods, Single-cell VAE (scVAE) [23] and scDeepSort

**Table 1. Summary of Experimental conditions.**

| Experiment conditions[*] | Pretrain | Eval 1 | Eval 2 | Eval 3 | Eval 4 | Eval 5 |
|---|---|---|---|---|---|---|
| Input max dimensions | 2048 | 2048 | 2048 | 2048 | 2048 | 2048 |
| Transformer Encoder Brocks | 6 | 6 | 6 | 6 | 6 | 6 |
| Attention Heads | 4 | 4 | 4 | 4 | 4 | 4 |
| Embedding dimensions | 256 | 256 | 256 | 256 | 256 | 256 |
| Activation function | SiLU | SiLU | SiLU | SiLU | SiLU | SiLU |
| Dropout rate | 0.02 | 0.02 | 0.02 | 0.02 | 0.02 | 0.02 |
| Epochs | 10 | 10 | 10 | 10 | 10 | 10 |
| Max learning rate | 0.001 | 5.00E-05 | 5.00E-05 | 5.00E-05 | 5.00E-05 | 5.00E-05 |
| Warm up steps | 10000 | 500 | 500 | 500 | 500 | 500 |
| Scheduler | cosine | cosine | cosine | cosine | cosine | cosine |
| Batch sizes | 12 | 12 | 12 | 12 | 12 | 12 |
| Optimizer | AdamW | AdamW | AdamW | AdamW | AdamW | AdamW |
| Weight decay | 0.001 | 0.001 | 0.001 | 0.001 | 0.001 | 0.001 |
| Fleeze Brocks | 0 | 0 | 6 | 0 | 0 | 0 |

[*]Each column represents the parameters used in the following experiments: Pretrain – pretraining of the mouse-Geneformer; Eval 1 – fine-tuning for cell type classification using mouse-Geneformer and conventional methods; Eval 2 – cell type classification using mouse-Geneformer with and without prior learning; Eval 3 – in silico perturbation experiments using mouse-Geneformer; Eval 4 – cell type classification after gene name conversion between human and mouse. Eval 5 – in silico perturbation experiments after gene name conversion between human and mouse.

[24] . The mouse data used for comparison comprised single-cell data from twelve organs of mice: urethra and prostate mixed dataset [25] (Gene Expression Omnibus (GEO): GSE145929), embryos [26] (Gene Expression Omnibus (GEO): GSE197353), and kidneys [27] (Gene Expression Omnibus (GSE): GSE190094), tongue [28], thymus [28], mammary gland [28], large intestine [28], limb muscle [28], spleen [28], heart [28], brain [28], and kidney [28] (Gene Expression Omnibus (GSE): GSE132042). These datasets were downloaded from CELLxGENE (https://cellxgene.cziscience.com/datasets). These data were not included in the mouse-Genecorpus-20M. To confirm the integrity of the data, we compared features using Euclidean distance, and the results indicated that no data was completely zero, confirming the absence of data leakage. Each method was employed to classify the data, and their accuracies are compared. For cell type classification with mouse-Geneformer, the model was fine-tuned for each organ data for this task. The fine-tuning conditions are detailed in Table 1. For cell type classification with scVAE, a method applying the deep generative model Gaussian-mixture VAE (GMVAE) [29] to single-cell RNA-seq analysis [23], we used the GMVAE with a three-layer neural network for the encoder, a Gaussian mixture distributions for the latent space, a three-layer neural network for the decoder, and a three-layer neural network for the class classification network. Additionally, the categorical distribution used in this GMVAE employed the Gumbel-Softmax [30,31]. For cell type classification with scDeepSort, a pretrained deep learning model utilizing weighted Graph Neural Networks (GNNs) for single-cell RNA-seq analysis [32], we employed its three-layer graph neural network architecture [24]. The data were randomly split into two groups, one for training (80%) and the other for testing, (20%). The evaluation metric for this experiment were accuracy and F1 score (a harmonic mean of precision and recall), which were calculated by using the 'accuracy_score' and 'f1_score' function from the scikit-learn library in Python, respectively, and the highest accuracy and F1 scores was recorded. Cell distribution was visualized using UMAP [33] performed using the scanpy library in Python.

## Cell type classification using mouse-Geneformer with and without prior learning

To evaluate the effect of pretraining, we compared the performance of pretrained mouse-Geneformer with non-pretrained one for the task of classification of cell types. The data used for evaluation included single-cell RNA-seq data from the mouse urethra and prostate mixed dataset [25] (Gene Expression Omnibus (GEO): GSE145929), embryos [26] (Gene Expression Omnibus (GEO): GSE197353), and kidneys [27] (Gene Expression Omnibus (GSE): GSE190094). These datasets were downloaded from CELLxGENE (https://cellxgene. cziscience.com/datasets) and were not included in the mouse-Genecorpus-20M. This embryo dataset is whole mouse embryo at embryonic age E9.5. These datasets were not included in mouse-Genecorpus-20M. The mouse-Geneformer models with and without pretraining were employed to classify the data, and their accuracies are compared. We compared the similarity between the data used for pretraining and the data used for evaluation by calculating the Euclidean distance of their features. As a result, no data was found to be completely zero, confirming the absence of data leakage. To perform cell type classification with mouse-Geneformer, the models are fine-tuned for each organ data for the cell type classification task. The fine-tuning conditions are presented in Table 1. The data splitting method, evaluation metrics, and cell distribution visualization methods are the same as those used in cell type classification experiments (see above).

## In silico perturbation experiments

The in silico gene manipulation experiment using Mouse-Geneformer applied perturbations to each gene to identify those that contribute significantly to the disease. The Mouse-Geneformer in silico perturbation experiment first normalized the gene expression data and then applied Rank Value Encoding to rank the expression of each gene, creating a ranked gene table. Then, Mouse-Geneformer was fine-tuned for disease classification tasks, ensuring that the disease classification accuracy on the test data exceeds 90%. Subsequently, the genes in the dataset were randomly perturbed multiple times, and inference was performed using the fine-tuned model. For "virtual" gene deletion in silico, rather than setting its expression value to 0, we completely removed the gene from the dataset. The rank values of genes with lower ranks than the deleted gene were each increased by one. Conversely, for "virtual" activating a gene in silico, the target gene was repositioned to have a rank value of 1, and the rank values of genes ranked higher than the activated gene were each decreased by one. Cosine similarity was used to quantify the distances between the perturbed cell state and a specific cell state. To analyze the effect of genetic perturbations, statistical significance was assessed using the Wilcoxon rank sum test (Mann-Whitney U test), following the approach used in the original human Geneformer study [10]. First, the gene expression table of each reference mouse cell is input into mouse-Geneformer, and an average feature vector representing the gene expression state of the cell was computed. Next, the gene expression table of a cell with a perturbed specific gene was input into mouse-Geneformer to generate a new set of feature vectors. The cosine similarity between these feature vectors and the reference average feature vector was then calculated to obtain the "distribution of cosine similarity under specific gene perturbation." To establish a control, a random gene in the gene table of each perturbed mouse cell was perturbed, and the feature vectors were computed in the same manner. The cosine similarity between this set of feature vectors and the reference average feature vector was then calculated, yielding the "distribution of cosine similarity under random gene perturbation." The Wilcoxon rank-sum test was performed on these two distributions to test the null hypothesis: "The effect of perturbing a specific

gene on cell state is comparable to that of perturbing a random gene." If the p-value was below 0.05, the null hypothesis was rejected, indicating that the perturbation of the specific gene has a statistically significant effect on the cell state. This statistical approach allowed for a quantitative assessment of the extent to which perturbation of a specific gene affects cell state compared to random perturbation, thereby enabling the prediction of genes potentially associated with disease.

To evaluate the effectiveness of in silico perturbation experiments using mouse-Geneformer, we compared the genes predicted in silico to those identified in vivo experiments. The data used for evaluation included single-cell data from diabetic nephropathy, UMOD nephropathy, and normal kidney data [27] (Gene Expression Omnibus (GSE): GSE190094), as well as data from cells with knocked out COP1 protein and normal cells [34] (Gene Expression Omnibus (GSE): GSE147559). These datasets were not included in mouse-Genecorpus-20M. To ensure the integrity of the data, we compared features using Euclidean distance, and the results indicated that no data was completely zero, confirming the absence of data leakage. To perform disease typing with mouse-Geneformer, the model was fine-tuned for each disease data for this task. Fine tuning conditions are shown in Table 1. Disease labels were assigned at the organ level: cells from diseased organs were labeled as diseased, while cells from normal organs were labeled as normal. The cells were then classified according to these labels. Only models that achieved disease subtype classification accuracies above 90.00% on the test data were used for evaluation. The data were randomly split into training (80%) and testing (20%) sets. For the in silico perturbation experiments for diabetic nephropathy, we perturbed the cell states of normal kidney cells to resemble those of diabetic nephropathy. For UMOD nephropathy experiments, we perturb the cell states of cells from UMOD nephropathy to resemble normal kidney cells. In COP1 KO experiments, we perturb the cell states of cells with COP1 knocked out to resemble normal cells.

## Gene name conversion between mouse and human for cross-species application

To convert human genes to mouse genes for cross-species application, we used the databases of Mouse Genome Informatics (MGI) [35] and the HUGO Gene Nomenclature Committee (HGNC) [36]. HGNC BioMart application (https://biomart.genenames.org/martform/#!/default/HGNC?datasets=hgnc_gene_mart_2024_03_26) serves the function to provide MGI IDs for mouse homologs of queried human genes. The MGI Batch Summary (https://www.informatics.jax.org/batch/summary) provides a feature to convert Ensembl IDs to corresponding mouse MGI IDs queried. Specifically, the conversion from human genes to mouse genes involved the following steps: 1) converting human Ensembl IDs to mouse MGI IDs using HGNC; 2) converting mouse MGI IDs to mouse Ensembl IDs and Gene Symbols using MGI. When converting human Ensembl IDs to mouse Ensembl IDs, we use a conversion table that ensures a one-to-one correspondence. In cases where the human Ensembl ID to the mouse homolog MGI ID and MGI ID to the mouse Ensembl ID don't have a one-to-one correspondence, the following steps are taken to ensure a one-to-one correspondence between human Ensembl IDs and mouse Ensembl IDs: 1) Sort the gene symbols corresponding to human Ensembl IDs in alphabetical order, and then map the last gene symbol to the mouse homolog MGI ID for the human Ensembl ID; 2) Map the mouse homolog MGI ID to the final mouse Ensembl ID in the output. After converting human Ensembl IDs to mouse Ensembl IDs, we obtain 18,269 genes. There are 910 human Ensembl IDs that have a one-to-many correspondence, and there are 687 mouse homolog MGI IDs that have a one-to-many correspondence.

## Human cell type classification using mouse-Geneformer

To explore whether mouse-Geneformer could be used for the analyses of other organisms, we analyzed three single-cell RNA-seq datasets derived from three human organs: breast [37] (Gene Expression Omnibus (GEO): GSE195665), thymus [38] (Gene Expression Omnibus (GEO): GSE144870), and cerebral cortex [39] (Brain Initiative Cell Census Network (BICCN): SCR_016152). These datasets were downloaded from CELLxGENE (https://cellxgene.czi-science.com/datasets) and were not included in Genecorpus-30M. To confirm the integrity of the data, we compared features using Euclidean distance, and the results indicated that no data was completely zero, confirming the absence of data leakage. To perform human cell type classification using mouse-Geneformer, human genes in the original datasets were converted to their mouse homologs as described in the previous section. The mouse-Geneformer and Geneformer were employed to classify each data, and their accuracies are compared. The Geneformer was a human-Geneformer model that we pre-trained using Genecorpus-30M (https://huggingface.co/datasets/ctheodoris/Genecorpus-30M). For cell type classification with mouse-Geneformer and Geneformer, models were fine-tuned for each organ data for this task. The fine-tuning conditions are detailed in Table 1. The procedures of fine-tuning and classification task were the same for both the mouse-Geneformer and human-Geneformer models. The data were randomly split into training (80%) and testing (20%) sets. Accuracy and F1 score were used as evaluation metrics. Cell distribution was visualized using UMAP [33] as described above.

## In silico perturbation of human data using mouse-Geneformer

To explore whether mouse-Geneformer can be used for the analyses of other organisms, we analyzed human single-cell RNA-seq data from myocardial infarction cells [40] (European Genome-phenome Archive: EGAS00001006330) and COVID-19 human blood cells [41] (Gene Expression Omnibus (GEO), European Genome-phenome Archive: GSE150728,GSE 155673,GSE150861,GSE149689,EGAS00001004571). These datasets were downloaded from CELLxGENE (https://cellxgene.cziscience.com/datasets). To perform in silico perturbation experiments on human cells using mouse-Geneformer, human genes in the original datasets were converted to their mouse homologs as described in the previous section. To perform disease typing with mouse-Geneformer, the model was fine-tuned for each disease data for this task. Fine tuning conditions are shown in Table 1. Only models achieving disease subtype classification accuracies above 90.00% on the test data were used for evaluation. Each data-set was randomly split into training (80%) and testing (20%) sets. The in silico perturbation experiments aimed to alter the cell expression profiles of normal cells to resemble abnormal cells and vice versa. For comparison, these human datasets were also analyzed using the original human-Geneformer model.

The number of cells, cell types, and types of diseases used in this experiment with single-cell data of mice are shown in S2 Table.

## Human cell type classification using mouse-Geneformer and scGPT in zero-shot mode

We analyzed three single-cell RNA-seq datasets derived from three human organs: breast [37] (Gene Expression Omnibus (GEO): GSE195665), thymus [38] (Gene Expression Omnibus (GEO): GSE144870), and cerebral cortex [39] (Brain Initiative Cell Census Network (BICCN): SCR_016152). These datasets were downloaded from CELLxGENE (https://cellxgene.czi-science.com/datasets) and were not included in Genecorpus-30M. To ensure data integrity, we assessed dataset features using Euclidean distance and confirmed that no dataset contained

entirely zero values, indicating the absence of data leakage. To perform human cell type classification using mouse-Geneformer, human gene names in the original datasets were converted to their mouse homologs as described in the previous section. To compare the classification performance of the mouse-Geneformer with human-native models, we also employed the original Geneformer (human-Geneformer) and scGPT, both of which were pre-trained on large-scale human scRNA-seq data. The human-Geneformer model was pre-trained using Genecorpus-30M (https://huggingface.co/datasets/ctheodoris/Genecorpus-30M). The human-scGPT model was obtained from the developer's repository (https://github.com/bowang-lab/scGPT) [42]. Zero-shot analysis was performed for all three models, where cell types were classified using the pre-trained models without any fine-tuning. The same procedures for zero-shot classification were applied across three models. Accuracy and F1 score were used as evaluation metrics to assess classification performance.

## Mouse cell type classification using human-Geneformer

To evaluate the applicability of the original human-Geneformer for mouse analysis, we analyzed twelve single-cell RNA-seq datasets derived from twelve organs of mice: urethra and prostate mixed dataset [25] (Gene Expression Omnibus (GEO): GSE145929), embryos [26] (Gene Expression Omnibus (GEO): GSE197353), and kidneys [27] (Gene Expression Omnibus (GSE): GSE190094), tongue [28], thymus [28], mammary gland [28], large intestine [28], limb muscle [28], spleen [28], heart [28], brain [28], kidney [28] (Gene Expression Omnibus (GSE): GSE132042). These datasets were downloaded from CELLxGENE (https://cellxgene.cziscience.com/datasets) and were not included in the mouse-Genecorpus-20M. To confirm the integrity of the data, we compared features using Euclidean distance, and the results indicated that no data was completely zero, confirming the absence of data leakage. To perform mouse cell type classification using Geneformer, mouse genes in the original datasets were converted to their human homologs as described in the previous section. We performed zero-shot classification on these datasets and compared their accuracy. Zero-shot classification involves classifying data without fine-tuning the pretrained model. Accuracy and F1 score were used as evaluation metrics.

## Mouse cell type classification using scGPT

We analyzed twelve single-cell RNA-seq datasets derived from twelve mouse organs, using the same datasets described in the section "Mouse cell type classification using human-Geneformer". To perform mouse cell type classification with scGPT [42], a model trained on human data, mouse gene names in the original datasets were converted to their human homologs, as described in the previous section. Zero-shot classification was then performed on these datasets following the procedures outlined in "Human cell type classification using mouse-Geneformer and scGPT in zero-shot mode". Classification performance was evaluated using accuracy and F1 score as metrics.

## Results

### Development of mouse-Geneformer

Theodoris et al. originally developed Geneformer, a context-aware, attention-based deep learning model, which was pretrained on large-scale transcriptome data comprising approximately 30 million human single-cell RNA-seq data [10]. While the original Geneformer was tailored for human transcriptome analysis, our study focuses on creating a mouse version, referred to mouse-Geneformer, from mouse transcriptome data. By leveraging the mouse-Geneformer, we aim to harness the capabilities of large-scale deep learning models to

enhance research involving mice, the most extensively studied model organism. The architecture and transfer learning strategy employed in developing mouse-Geneformer are the same as those of the original human Geneformer, with minor modifications. The primary but critical difference lies in the input corpus, which was constructed from mouse single-cell RNA-seq data.

We constructed a large-scale dataset of mouse single-cell RNA-seq data, termed mouse-Genecorpus-20M (Fig 2A). The mouse-Genecorpus-20M comprises approximately 21 million raw single-cell data from healthy mice, encompassing a wide variety of organs and developmental stages. The breakdown of the dataset is shown in Fig 2B,2C and 2D and detailed information is provided in S1 Table. It is noteworthy that mouse-Genecorpus include types of samples that are difficult to obtain from humans due to ethical or technical constraints (e.g., embryos). Since mouse-Genecorpus-20M is intended for learning the gene network of normal mice, we excluded single-cell data with high mutation loads, such as cancer cells or immortalized cell lines, that could restructure gene networks. We also omitted low-quality RNA-seq data derived from doublets and damaged cells. After the filtration, single-cell transcriptome data from 20,630,028 cells remained, constituting mouse-Genecorpus-20M (Fig 2A).

The transcriptome of mouse-Genecorpus-20M was modeled using the rank value encoding method and then processed through six transformer encoder units, as employed in the original Geneformer. This process yielded "cell sequences" represented by gene names as tokens. Pretraining was conducted using a masked language model under the conditions shown in Table 1. We made several modifications to the original Geneformer parameters: 1) We changed the activation function from ReLU to SiLU (Sigmoid-weighted Linear Unit, also known as Swish) because SiLU provides smoother gradients than ReLU. It also provides gradients when the input values are less than or equal to 0, and increases the gradient when the input values are greater than or equal to 1. Therefore, it is less prone to the vanishing gradient problem and can improve the performance in deep neural network models. 2) We increased the number of epochs from three to ten to allow the number of training times on the model, reduced learning loss and improved performance. 3) In comparing linear and cosine scheduling modes for the learning rate scheduler, we found that pre-training with the cosine schedule showed better performance by reducing learning loss more effectively. Therefore, we chose the cosine schedule. As a result of these procedures, the mouse-Geneformer was constructed.

## Cell type classification: pretraining is effective

To evaluate the effect of pretraining, we compared the cell type classification results of the mouse-Geneformer with and without pretraining, as shown in Table 2. We conducted three experiments using samples from 1) the prostate gland and urethra, 2) embryo that is whole mouse embryo at embryonic age E9.5, and 3) kidney. Our findings indicated that the classification accuracy of the mouse-Geneformer with pretraining was higher than that without

**Table 2. Cell type classification results with and without prior learning.**

| Tissue | # Cell types | w/ prior | | w/o prior | |
|---|---|---|---|---|---|
| | | Acc | F1 | Acc | F1 |
| Prostate gland & Urethra | 7 | 94.70 | 90.25 | 93.73 | 88.71 |
| Embryo | 9 | 75.24 | 67.09 | 66.56 | 45.42 |
| Kidney-1 | 10 | 79.02 | 38.65 | 78.91 | 38.54 |

Cell type classification performance across different tissues was evaluated using models with prior learning (w/ prior) and without prior learning (w/o prior). Accuracy (Acc) and F1 score (F1) are presented as percentages (%) and were used as evaluation metrics.

pretraining in all cases. The maximum difference in classification accuracy was 8.22% for the embryo data, while the improvements of the other cases were marginal. UMAP visualization of the cell distribution in the embryo data demonstrated that the pretrained mouse-Geneformer depicted more discrete cell clusters compared to the non-pretrained version as shown in Fig 3A and 3B. For example, primitive erythrocytes and cardiac valve cells are distributed among multiple clusters in the non-pretrained mouse-Geneformer classification, whereas in the pretrained version, they formed distinct clusters. These results suggests that pretraining is effective in enhancing mouse-Geneformer, leading to improved classification accuracy.

## Cell type classification: mouse-Geneformer is robust and outperforms conventional methods

We investigated the performance of mouse-Geneformer in cell-type classification across various organs. Twelve experiments were conducted using different organs to compare the accuracy of mouse-Geneformer with conventional methods such as scDeepSort and scVAE (Table 3). Our findings demonstrated that mouse-Geneformer greatly improves the classification accuracy of cell types in all cases, achieving an average accuracy of 96.73%. In comparison, the GNN-based method scDeepSort and Autoencoder-based method scVAE showed accuracy scores of 66.34% and 72.95%, respectively. Notably, mouse-Geneformer constantly achieved classification accuracies exceeding 93%, regardless of target organs or the number

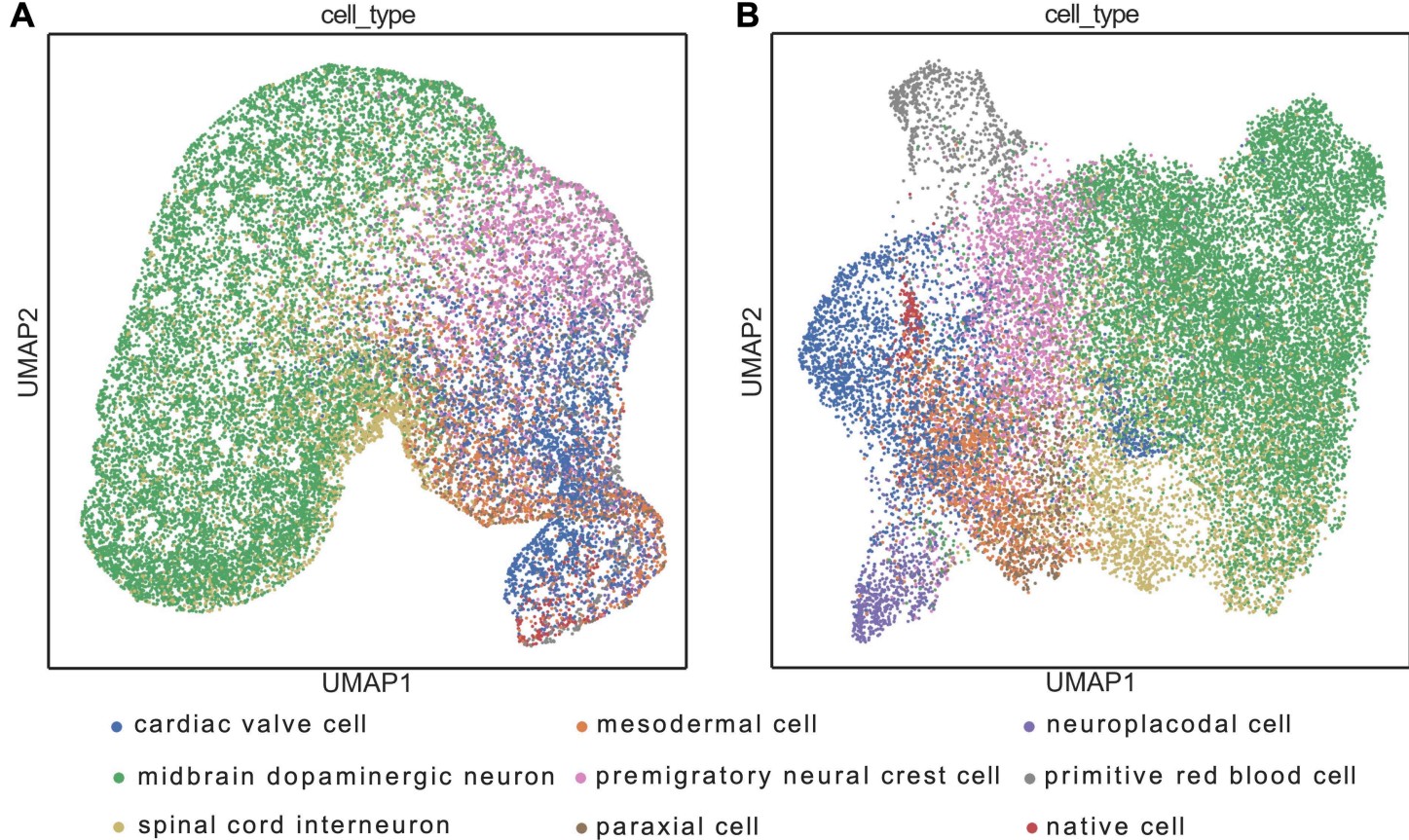

**Fig 3. UMAP visualization of mouse-Geneformer cell embeddings for embryonic cells.** (A) UMAP plot of cell embeddings generated by mouse-Geneformer without prior learning. (B) UMAP plot of cell embeddings generated by mouse-Geneformer with prior learning. Cell type annotations are derived from the original study [26].

**Table 3. Cell type classification results using the mouse-Geneformer and two conventional methods.**

| Tissue | #Cell types | Mouse-Geneformer | | scDeepSort | | Single-cell VAE | |
|---|---|---|---|---|---|---|---|
| | | Acc | F1 | Acc | F1 | Acc | F1 |
| Prostate gland & Urethra | 7 | 94.70 | 90.25 | 80.64 | 40.72 | 73.74 | 39.66 |
| Embryo | 9 | 75.24 | 67.09 | 49.44 | 4.72 | 52.22 | 11.55 |
| Kidney-1 | 10 | 79.02 | 38.65 | 64.23 | 18.63 | 67.68 | 33.56 |
| Tongue | 3 | 94.87 | 96.13 | 76.79 | 19.98 | 80.44 | 20.55 |
| Thymus | 6 | 96.97 | 96.34 | 54.94 | 13.17 | 74.95 | 29.33 |
| Mammary Gland | 7 | 99.02 | 98.57 | 48.02 | 20.33 | 79.54 | 36.86 |
| Large Intestine | 7 | 93.08 | 92.08 | 58.46 | 20.95 | 59.00 | 24.41 |
| Limb Muscle | 9 | 99.52 | 98.69 | 90.82 | 57.77 | 79.58 | 66.44 |
| Spleen | 10 | 98.70 | 97.27 | 81.01 | 36.74 | 76.47 | 26.62 |
| Heart | 11 | 97.82 | 95.86 | 79.55 | 23.29 | 79.42 | 20.39 |
| Brain | 15 | 96.92 | 88.01 | 58.53 | 10.08 | 76.19 | 49.83 |
| Kidney-2 | 18 | 94.88 | 90.86 | 57.60 | 22.44 | 56.25 | 20.73 |

Accuracy (Acc) and F1 score (F1) are presented as percentages (%) and were used as evaluation metrics.

of cell types. UMAP visualization of the tongue and limb muscle samples further supported the good performance of mouse-Geneformer in cell-type annotation (Fig 4A and 4B). Each cell type was clearly separated, although the overlap of the distribution of Langerhans cells with epithelial basal cells posed a challenge in the tongue dataset. These results indicate that mouse-Geneformer is more accurate than existing methods and robust across different organs and varying cellular complexity. This suggests that mouse-Geneformer can be applicable effectively to a wide variety of mouse organs.

## In silico perturbation experiments

We next investigated the applicability of mouse-Geneformer to in silico perturbation experiments for mice. In silico perturbation experiments simulate genetic manipulation experiments on computers, allowing us to mimic gene deletions or activations by editing the gene list within each cell. This enables the prediction of how these mutations will impact the gene network and elucidate the functions of the mutated genes. Notably, this approach provides a powerful method for candidate gene screening, as it allows for the repeated simulation of experiments for multiple target genes of interest, or even all mouse genes, in silico. By comparing the effects of these perturbations, we can identify the most influential genes. Detailed information about in silico perturbation experiment is provided in Methods. We here analyzed three disease models using mouse-Geneformer.

**In silico perturbation experiment in diabetic kidney disease.** We analyzed diabetic nephropathy using a set of single-cell transcriptome data collected from disease kidney exhibiting diabetes and from normal kidneys as a control. Detailed information about data and fine turning are provided in Methods. The UMAP visualization of the cell distribution shown in Fig 5A showed a clear separation between normal kidney cells and disease cells.

Subsequently, we conducted in silico perturbation experiments by randomly and repeatedly choosing target genes. Detailed information about in silico perturbation experiment is provided in Methods. The analysis revealed that deleting the gene *Slc12a3* from normal kidney cells brought the cells closest to those of diabetic kidney disease, with a cosine similarity of 0.018. This in silico outcome is consistent with observations from in vivo experiments,

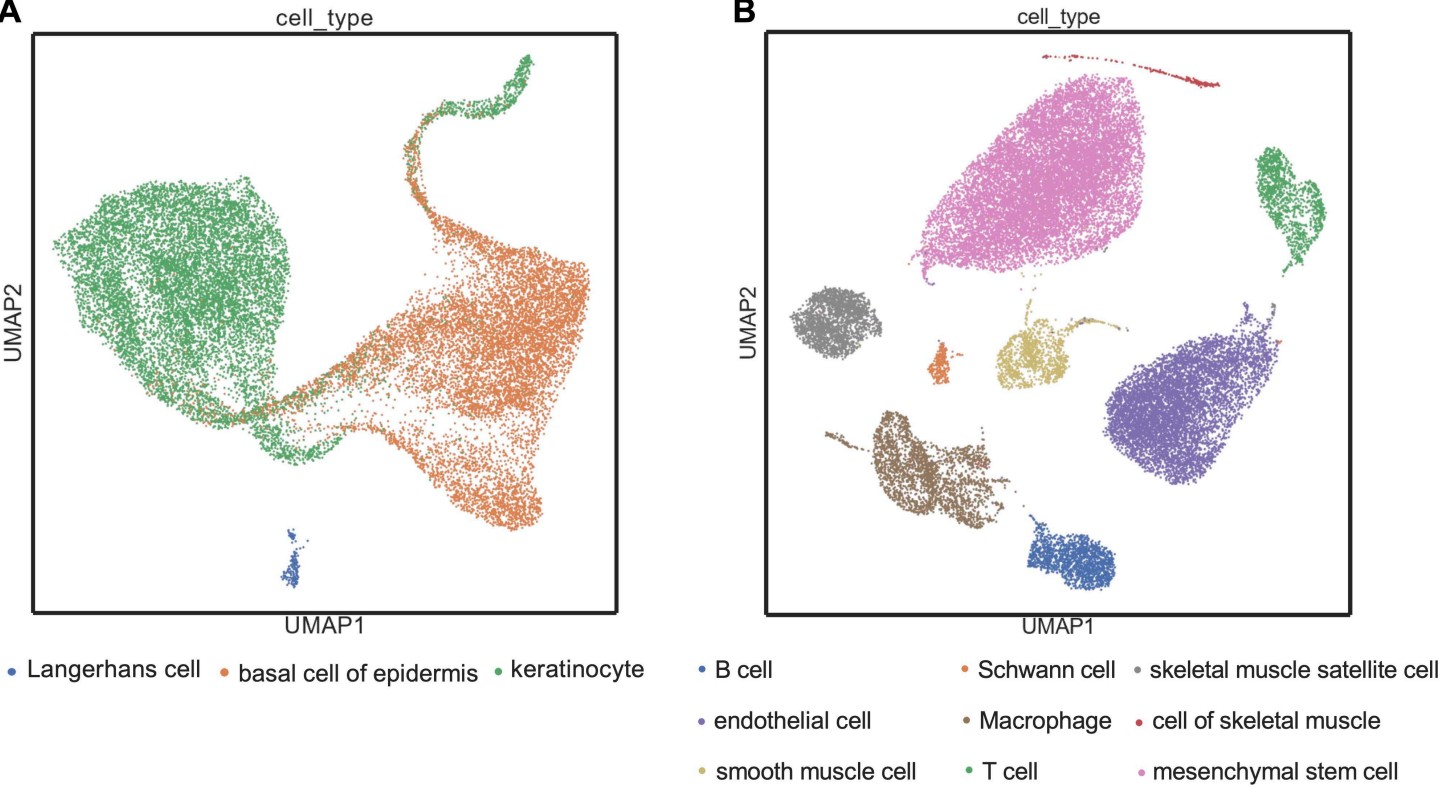

**Fig 4. UMAP visualization of mouse-Geneformer cell embeddings.** (A) UMAP plot of cell embeddings generated by mouse-Geneformer for tongue cells. (B) UMAP plot of cell embeddings generated by mouse-Geneformer for limbs cells. Cell type annotations are derived from the original studies [28].

where granular cells of the glomerular apparatus suffered from diabetic kidney disease showed diminished expression of *Slc12a3* [27].

**In silico perturbation experiment in UMOD kidney disease.** Following the similar approach, we analyzed UMOD kidney disease. We modeled the cell status of UMOD kidney disease by fine-tuning our mouse-Geneformer using the single-cell transcriptome data (Fig 5A). In silico perturbation of the disease cells by altering gene expressions revealed that deleting the gene *Slc35b1*, abundantly expressed in cells of UMOD kidney disease, had the greatest impact in moving the disease cell status closer to normal kidney cells. In in vivo experiments, it has been reported that the accumulation of mutant UMOD protein in the kidney alters the Unfolded Protein Response (UPR), leading to changes in the expression of 77 genes, including *Slc35b1* and *Slc3a2* that activate the UPR [27]. These observations indicate that our in silico perturbation experiments for UMOD kidney disease successfully identified one of the two genes validated by the in vivo experiment.

**In silico perturbation experiment in COP1 KO microglia.** Using a similar approach, we analyzed a model of neuroinflammation disease. We utilized single-cell RNA-seq data of microglia cells with COP1 knocked out, a gene known for suppressing neuroinflammation, and normal cells [34]. Detailed information about data and fine turning are provided in Methods. Fig 5B exhibits complete separation between normal cells (Cop1 WT) and COP1 knocked-out cells (Cop1 KO). Then, we conducted in silico perturbation experiments. Deletion of *Apoe* gene from COP1 KO cells demonstrated the closest resemblance to normal cells, with a cosine similarity of 0.36. Additionally, deleting the genes *Fth1*, *Itgax*, and *Cst7* also showed significant impacts in altering the cells closer to normal cells. These observations are

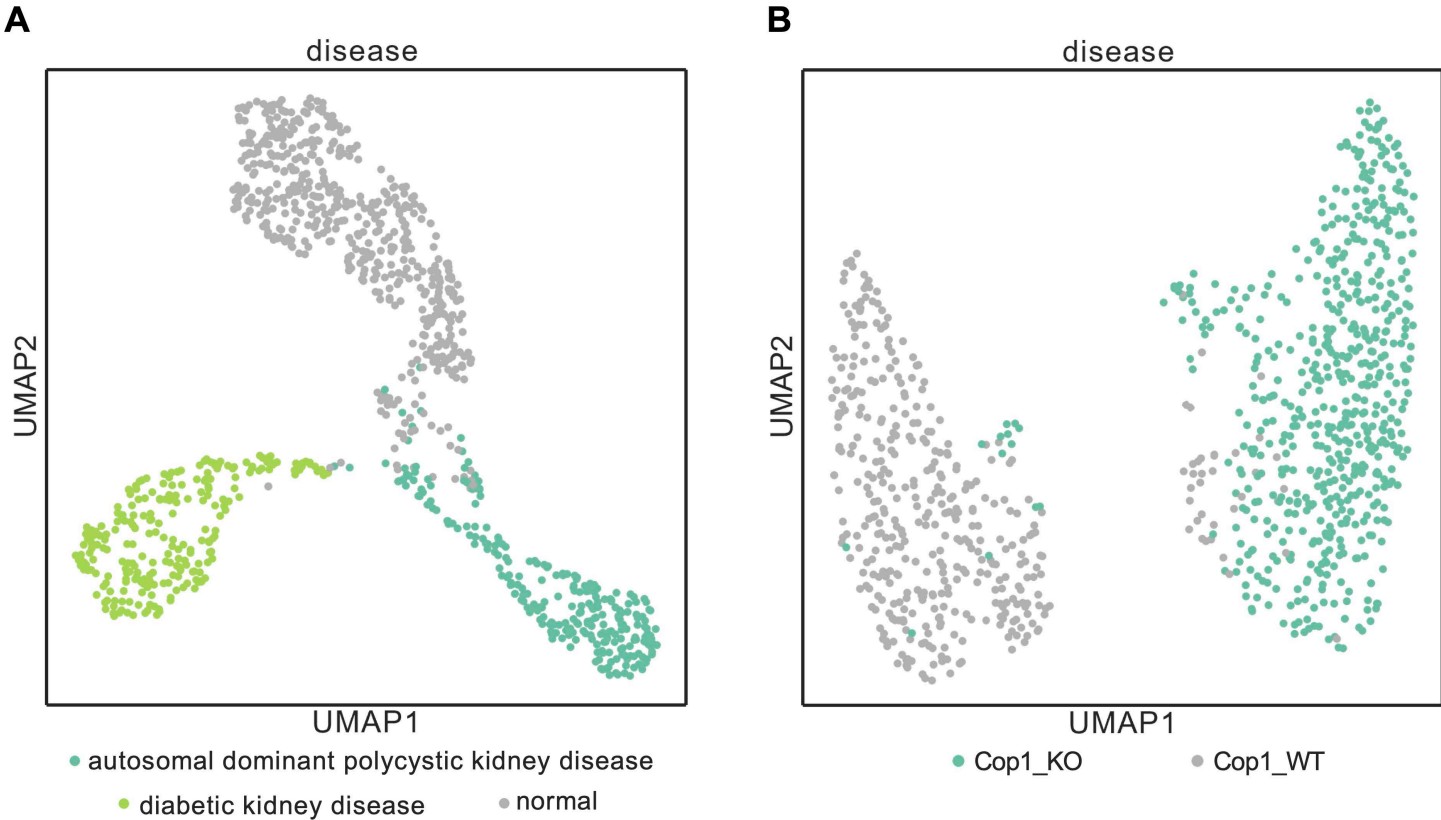

**Fig 5. UMAP visualization of mouse-Geneformer cell embeddings for cells of disease models.** (A) UMAP plot of cell embeddings generated by mouse-Geneformer for cells derived from three kidney disease models. (B) UMAP plot of cell embeddings generated by mouse-Geneformer for cells derived from normal microglia cells (Cop1 WT) and COP1 knocked-out cells (Cop1 KO).

consistent with in vivo experiments, which reported that neuroinflammation occurs due to the accumulation of c/EBPβ in the brain when COP1 is knocked out, and that knocking out COP1 increases the expression levels of genes *Apoe* and *Fth1* and affects the expression levels of neurodegeneration-related genes *Itgax* and *Cst7* [34].

Altogether, in silico perturbation experiments, following the fine-tuning of mouse-Geneformer model using single-cell transcriptome data of target disease model, effectively recapture the results of in vivo experiments.

## Cross-species application of mouse-Geneformer through orthologous gene name conversion

We explored whether mouse-Geneformer could be used for the analyses of other organisms. If successful, this would allow non-model mammals, for which it is difficult, costly, and technically challenging to obtain sufficient single-cell transcriptome data to construct species-specific Geneformer models, to benefit from transfer learning using the mouse gene network model. Given the core metabolic and physiological features are conserved among closely related species, we hypothesize that the core genetic architectures would also be conserved. Therefore, mouse-Geneformer could potentially elucidate a significant portion of genetic architectures beyond mice. Even humans could benefit from mouse-Geneformer, despite the existence of the original Geneformer. This is because mouse-Genecorpus can include types of samples that are difficult to obtain from humans due to ethical or technical constraints.

As a proof of concept for the cross-species application of mouse-Geneformer, we investigated its applicability to human transcriptome analysis. The outline of the procedure was as follows: First, to analyze human genes in mouse-Geneformer, each human gene name was converted to mouse ortholog based on the ortholog table. We then fine-tuned the mouse-Geneformer model using human transcriptome data converted to mouse ortholog. Using this fine-tuned model, we conducted in silico perturbation experiments.

**Human cell type classification using mouse-Geneformer.** We conducted human cell type classification using human single-cell RNAseq data with mouse-Geneformer. Three experiments were conducted using different human organs. Detailed information about how to convert gene and fine tuning is provided in Methods. For comparison, we also analyzed the human data using original human-Geneformer model.

The cell type classification results are shown in Table 4. The results demonstrated that mouse-Geneformer (with fine-tuning) accurately classified human cell types based on the human transcriptome data for all cases, achieving accuracy scores of 95.59%, 99.97% and 87.82% for human thymus, cerebral cortex, and breast, respectively. In addition, the classification accuracy of the ortholog-converted mouse-Geneformer for human data was nearly equivalent to that of the original human-Geneformer, with marginal differences ranging from 0.01 to 0.30%. F1 scores were also comparative each other. These data indicate that mouse-Geneformer is effective for cell type classification of human cells and suggests that the Geneformer model can work effectively across species following orthology-based data conversion.

**In silico perturbation of human data using mouse-Geneformer 1: myocardial infarction.** Encouraged by the success of applying mouse-Geneformer to human cell type classification, we next investigated its utility for in silico perturbation experiments on human data. We analyzed human single-cell RNA-seq data from myocardial infarction cells to evaluate the model's effectiveness. The original human-Geneformer predicted that activation of *NPPB* and *ANKRD1*, as well as deletion of *MYH7*, could drive normal heart cells towards myocardial infarction state. The mouse-Geneformer similarly indicated that activation of *Ankrd1* and deletion of *Myh7* could lead to a comparable transition from normal heart cells towards myocardial infarction cells. These result aligns with findings from single-cell RNA sequencing analyses of human myocardial infarction and normal heart cells [40]. Conversely, human-Geneformer model predicted that deleting *ANKRD1* and *NPPB* from myocardial infarction cells would revert them to a more normal cellular state. Similarly, the mouse-Geneformer model predicted that deletion of genes *Nppb*, *Ankrd1*, and *Myh7* from myocardial infarction cells would also lead to a transition towards a normal state. Thus, these results demonstrate that mouse-Geneformer can effectively perform in silico perturbation experiments on human disease models like myocardial infarction.

**Table 4. Human cell type classification using mouse-Geneformer via ortholog-based gene name conversion, compared to native human models (human-Geneformer and scGPT).**

| Tissue | #cell types | Mouse-Geneformer (Zero-shot) | | Mouse-Geneformer (Fine-tuning) | | Human-Geneformer (Zero-shot) | | Human-Geneformer (Fine-tuning) | | scGPT | |
|---|---|---|---|---|---|---|---|---|---|---|---|
| | | Acc | F1 | Acc | F1 | Acc | F1 | Acc | F1 | Acc | F1 |
| h/ Thymus | 4 | 82.74 | 48.57 | 95.59 | 87.81 | 87.27 | 74.48 | 95.44 | 87.79 | 92.04 | 82.37 |
| h/ Celebral Cortex | 6 | 89.17 | 68.79 | 99.97 | 99.95 | 99.49 | 98.41 | 99.98 | 99.97 | 99.98 | 99.97 |
| h/ Breast | 12 | 34.63 | 17.35 | 87.82 | 80.44 | 55.97 | 43.99 | 88.12 | 82.89 | 82.43 | 74.89 |

Accuracy (Acc) and F1 score (F1) are shown as percentages (%) and were used as evaluation metrics. The prefix "h/" indicates human-derived datasets.

**In silico perturbation of human data using mouse-Geneformer 2: COVID-19 human blood.** Next, we investigated the effectiveness of mouse-Geneformer in analyzing a human-specific disease. We conducted perturbation experiments on COVID-19 human blood cells using both human-Geneformer and mouse-Geneformer for comparison. Notably, SARS-CoV-2, the coronavirus responsible for COVID-19, does not naturally infect wild-type laboratory mice [43,44]. Using the human Geneformer, we predicted that the activation of genes *CCR4*, *IL6*, and *CCR20* in normal human blood cells would lead to a transition towards a state resembling SARS-CoV-2 infection. This prediction is consistent with findings from single-cell RNA sequencing analysis of normal blood and COVID-19 blood [41]. The mouse-Geneformer predicted that the activation of genes *Cxcl3* and *Ccr4* would induce a similar transition towards COVID-19-infected cells. Conversely, in silico perturbation of COVID-19 human blood cells using human-Geneformer model predicted that deletion of genes *CXCL2*, *IFITM3*, and *CCL20* would revert the COVID-19 cells towards a normal state. The mouse-Geneformer model predicted that deleting genes *Ccr4* and *Il6* would lead to a transition towards normal cells. These results suggest that both human-Geneformer and mouse-Geneformer predict the involvement of similar genes, such as *CCR4* and *IL6*, in the process of SARS-CoV-2 infection in human blood cells. However, the overlap of the genes predicted by both models was not large. This may indicate limitations in cross-species application of Geneformer model for species-specific traits. Thus, while mouse-Geneformer demonstrates potential in cross-species application for understanding human diseases like COVID-19, it also highlights the importance of species-specific models for capturing the full complexity of disease mechanisms.

**Mouse cell type classification using human Geneformer.** Given that the human transcriptome can be effectively analyzed using mouse-Geneformer, we sought to evaluate the reverse scenario: whether the original human Geneformer could classify mouse cell types using mouse single-cell RNA sequencing data after homology-based gene name conversion. The results of this analysis are presented in Table 5. This demonstrated that human Geneformer successfully classified mouse cell types based on mouse transcriptome data across all cases, achieving performance levels comparable to those of mouse-Geneformer.

**Table 5. Mouse cell type classification using human-based models (human-Geneformer and scGPT) via ortholog-based gene name conversion.**

| Tissue | #cell types | Mouse-Geneformer | | Human-Geneformer | | scGPT | |
|---|---|---|---|---|---|---|---|
| | | Acc | F1 | Acc | F1 | Acc | F1 |
| m/Prostate gland & Urethra | 7 | 86.15 | 82.34 | 85.67 | 82.59 | 91.42 | 87.70 |
| m/Embryo | 9 | 53.41 | 33.36 | 55.90 | 35.20 | 63.99 | 49.13 |
| m/Kidney-1 | 10 | 71.19 | 31.35 | 64.22 | 33.24 | 59.89 | 30.59 |
| m/Tongue | 3 | 81.90 | 86.58 | 84.11 | 87.51 | 81.10 | 86.67 |
| m/Thymus | 6 | 83.96 | 81.60 | 86.12 | 81.78 | 94.49 | 91.99 |
| m/Mammary Gland | 7 | 96.77 | 95.99 | 94.53 | 93.58 | 97.38 | 96.27 |
| m/Large Intestine | 7 | 66.48 | 67.39 | 67.93 | 69.01 | 89.77 | 88.37 |
| m/Limb Muscle | 9 | 96.53 | 92.74 | 95.59 | 91.63 | 99.23 | 98.55 |
| m/Spleen | 10 | 92.98 | 92.00 | 93.97 | 91.63 | 97.65 | 95.86 |
| m/Heart | 11 | 88.24 | 81.45 | 88.91 | 79.84 | 95.91 | 88.29 |
| m/Brain | 15 | 86.56 | 75.18 | 84.95 | 67.73 | 96.08 | 86.51 |
| m/Kidney-2 | 18 | 85.84 | 81.14 | 76.87 | 67.40 | 82.00 | 67.22 |

Mouse-Geneformer is included for comparison. Accuracy (Acc) and F1 score (F1) are reported as percentages (%) and were used as evaluation metrics. All results were obtained in a zero-shot setting. The prefix "m/" indicates mouse-derived datasets.

This indicates that a homology-based approach is effective for cross-species applications of the Geneformer architecture between mouse and human. Interestingly, tasks involving cell type classification with a higher number of classes showed reduced performance in the cross-species application of Geneformer. This effect was particularly evident in brain data, which included 15 cell types. This suggests that more complex tasks require native transcriptome data for high-resolution classification, while simpler tasks, such as classification involving fewer cell types, can be effectively analyzed using a cross-species approach. Notably, approximately 15% of mouse genes could not be assigned to human orthologs, likely resulting in the loss of information related to mouse-specific genetic regulation. This limitation may have contributed to the lower performance observed in more complex classification tasks, further underscoring the importance of developing Geneformer models tailored to the target species.

**Cross-species approach using human-based omics models and comparison with mouse-Geneformer.**  The success of cross-species strategy to enable human Geneformer to analyze mouse data demonstrates the potential to adapt other human-centric tools for mouse omics data. Recently, advanced deep-learning methods have been employed to model human single-cell omics; among them, scGPT [42] stands out as a state-of-the-art architecture. scGPT is a large language model (LLM) framework based on a GPT-style transformer, trained on over 33 million human cells. While scGPT has shown exceptional performance in downstream tasks such as cell type annotation, data integration, and discovery within human single-cell analyses, it cannot directly handle mouse data because it was developed only on human transcriptomics. To address this limitation, we applied the same orthologous gene conversion strategy as we used for Geneformer, transforming mouse gene data into human orthologs to enable scGPT analysis. Using this approach, we classified mouse cell types from scRNA-seq data with scGPT after orthologous gene conversion (Table 5). The results show that scGPT successfully classified mouse cell types, even under zero-shot learning conditions.

When we compared its performance against our mouse-Geneformer model, scGPT outperformed mouse-Geneformer in nine out of twelve classification tasks. The superior performance of scGPT can likely be attributed to its GPT-based architecture, which has demonstrated outstanding capabilities in natural language processing, exemplified by tools like ChatGPT, while Geneformer relies on a BERT-based architecture.

## Discussion

Geneformer is an innovative context-aware, attention-based deep learning model pretrained on large-scale human single-cell RNA-seq data. It enables fine-tuning for a vast array of downstream tasks with limited task-specific data. Building on the successful development of Geneformer using human scRNA-seq data, we here developed a mouse version of Geneformer using mouse scRNA-seq data. The architecture and transfer learning strategy of the original human version were followed with minor modifications. The evaluation of our mouse-Geneformer indicated successful development, as fine-tuning for downstream tasks improved the accuracy of cell type classification in mouse data, and in silico simulations of gene manipulation in mouse disease models detected genes identified in in vivo experiments. This suggests that the architecture of the original human Geneformer, including key components such as rank value encoding, is robust and applicable to species beyond human. It is expected that this strategy and architecture can be applied to any species to build species-specific Geneformer models, provided that large-scale transcriptome data are available.

The mouse, *Mus musculus*, is the foremost mammalian model for studying human biology and disease. Extensive knowledge about mouse physiology, anatomy, and genetics, along with well-developed methods for genetic manipulation – such as creating transgenic, knockout,

and knockin animals – has positioned the mouse as a crucial model in various biological and medical fields. In this context, the mouse-Geneformer developed in this study would greatly benefit mouse studies. We constructed a large-scale dataset of mouse single-cell RNA-seq data, termed mouse-Genecorpus-20M, comprising approximately 21 million single-cell RNA-seq profiles from healthy mice, encompassing a wide variety of organs and developmental stages. Pretrained with this dataset, mouse-Geneformer gained a fundamental understanding of the genetic network dynamics of the mouse transcriptome. By leveraging the prior knowledge, the accuracy of cell type classification with mouse-Geneformer has significantly improved compared to traditional methods. Furthermore, in silico perturbation experiments using mouse-Geneformer successfully identified disease-causing genes that have been validated in in vivo experiments. Thus, the mouse-Geneformer not only enhances our ability to understand the genetic network of mice but also enables in silico screening of key genetic factors in disease models. Specifically, in silico prediction using mouse-Geneformer can help prioritize the genes to analyze before conducting animal experiments, thereby avoiding *ad hoc* gene knockouts, saving time, and reducing the need for sacrificing animals.

We found that the mouse-Geneformer can be used for the analyses of the other animal species in a cross-species manner. In this study, after the ortholog-based gene name conversion, the analysis of human scRNA-seq data using mouse-Geneformer followed by the fine-tuning with human data achieved cell type classification accuracy comparable to that obtained using the original human Geneformer. Also, mouse-Geneformer effectively performed in silico perturbation experiments on human disease models of myocardial infarction. Given the core metabolic and physiological features are conserved among mammals, the core genetic architectures should also be conserved, thereby the mouse-Geneformer worked for human transcriptome. Despite the existence of the original Geneformer tailored for human, human research could benefit from mouse-Geneformer. This is because mouse-Genecorpus can include types of samples that are ethically or technically inaccessible for humans, such as embryonic tissues and certain disease models. As the amount and variety of mouse scRNA-seq data continues to increase, the inclusion of additional datasets into the current mouse-Geneorpus-20M to create an expanded mouse-Genecorpus will enable mouse-Geneformer to learn more accurate gene networks. This enhanced model could serve as a reference not only for mouse but also human studies.

We demonstrated that a cross-species approach using homology-based gene name conversion is effective in three scenarios: (1) analyzing human data with mouse-Geneformer, (2) analyzing mouse data with human-Geneformer, and (3) analyzing mouse data with scGPT, a model built upon human transcriptome data. Despite these successful applications, it is important to acknowledge the limitations of the homology-based cross-species strategy, as gene homology alone cannot fully account for the functional and regulatory differences between species. Even when genes are homologous, their regulatory mechanisms, spatio-temporal expression patterns, and roles in biological pathways can vary significantly across species. In fact, we observed that mouse-Geneformer failed to fully predict the outcome of perturbation experiments on human COVID-19 blood cell data, whereas the original human Geneformer predicted more accurately. Note that, SARS-CoV-2, the coronavirus responsible for COVID-19, does not naturally infect wild-type laboratory mice, highlighting the species-specific differences in viral susceptibility. On the other hand, we predict that housekeeping metabolic processes involving highly conserved genes can be effectively analyzed across species. It is also possible that lineage-specific phenotypes are influenced by complex genetic networks that integrate both conserved core pathways and species-specific regulatory components. In such cases, even lineage-specific traits may be partially analyzed using a cross-species approach, with conserved genetic elements contributing to the predictive accuracy of

the model. Additionally, we observed that cell type classification tasks with a higher number of classes showed reduced performance in cross-species applications of Geneformer. This suggests that more complex tasks require native transcriptome data for high-resolution classification, as cross-species models may lack the species-specific information necessary to achieve the same level of accuracy. Further research is needed to explore the full potential and limitations of cross-species applications of Geneformer models, particularly in understanding how conserved and species-specific gene networks interact to influence biological processes across different organisms.

The cross-species application of Geneformer holds great potential for the analysis of non-model organisms, for which it is difficult, costly, and technically challenging to obtain sufficient single-cell transcriptome data to construct species-specific Geneformer models. However, since we investigated only a single combination of cross-species application between human and mouse, it remains unclear how closely related species can be analyzed with Geneformer models in a cross-species manner. Future research should explore the utility of cross-species applications across various evolutionary distances, such as within or across families, orders, or even phyla. It is reasonable to predict that the greater the evolutionary distance between species, the more challenging these applications will become. This is likely due to several factors, including reduced conservation of genetic regulatory networks, diminished functional conservation of homologous genes, and technical difficulties in accurately identifying orthologs across distantly related species. The success of such applications may also depend on the traits or genes of interest. We predict that housekeeping metabolic processes involving conserved genes can be analyzed across species, whereas lineage-specific traits may not be as effectively analyzed. Importantly, the cross-species approach is not limited to Geneformer but is a versatile strategy that can be applied to other large-scale omics models. In fact, we demonstrated the successful use of human-based scGPT for mouse data, further supporting the feasibility of this approach. Looking ahead, it is inevitable that more advanced algorithms and tools will be developed to model large-scale omics data for human and mouse. These tools will likely be applicable to other species of interest, enabling researchers to leverage existing resources in a cross-species manner to address a wide range of biological questions.

## Supporting information

**S1 Table. Detailed information of mosue-Genecorpus-20M.**
(XLSX)

**S2 Table. Overview of data using evaluation experiments.**
(XLSX)

## Acknowledgments

We are grateful to all members of the Yamashita Laboratory and the Fujiyoshi Laboratory for their assistance, advice, consultation, and cooperation in obtaining experimental data for this research.

## Author contributions

**Conceptualization:** Keita Ito, Tsubasa Hirakawa, Shuji Shigenobu, Hironobu Fujiyoshi, Takayoshi Yamashita.

**Data curation:** Keita Ito, Tsubasa Hirakawa, Shuji Shigenobu.

**Formal analysis:** Keita Ito.

**Funding acquisition:** Hironobu Fujiyoshi.

**Investigation:** Keita Ito, Tsubasa Hirakawa, Shuji Shigenobu, Hironobu Fujiyoshi, Takayoshi Yamashita.

**Methodology:** Keita Ito, Tsubasa Hirakawa, Shuji Shigenobu, Hironobu Fujiyoshi, Takayoshi Yamashita.

**Project administration:** Shuji Shigenobu, Hironobu Fujiyoshi, Takayoshi Yamashita.

**Resources:** Tsubasa Hirakawa, Shuji Shigenobu, Hironobu Fujiyoshi, Takayoshi Yamashita.

**Software:** Keita Ito, Tsubasa Hirakawa, Takayoshi Yamashita.

**Supervision:** Tsubasa Hirakawa, Shuji Shigenobu, Hironobu Fujiyoshi, Takayoshi Yamashita.

**Validation:** Keita Ito, Shuji Shigenobu.

**Visualization:** Keita Ito.

**Writing – original draft:** Keita Ito, Tsubasa Hirakawa, Shuji Shigenobu, Hironobu Fujiyoshi, Takayoshi Yamashita.

**Writing – review & editing:** Keita Ito, Tsubasa Hirakawa, Shuji Shigenobu, Hironobu Fujiyoshi, Takayoshi Yamashita.

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
