## [Decision Letter · Decision Letter 0]

28 Oct 2024

PGENETICS-D-24-01035Mouse-Geneformer: A Deep Learning Model for Mouse Single-Cell Transcriptome and Its Cross-Species UtilityPLOS Genetics Dear Dr. Shigenobu, Thank you for submitting your manuscript to PLOS Genetics. After careful consideration, we feel that it has merit but does not fully meet PLOS Genetics's publication criteria as it currently stands. Therefore, we invite you to submit a revised version of the manuscript that addresses the points raised during the review process. Please submit your revised manuscript within 60 days Dec 27 2024 11:59PM. If you will need more time than this to complete your revisions, please reply to this message or contact the journal office at plosgenetics@plos.org. Please include the following items when submitting your revised manuscript:* A rebuttal letter that responds to each point raised by the editor and reviewer(s). You should upload this letter as a separate file labeled 'Response to Reviewers '. This file does not need to include responses to any formatting updates and technical items listed in the 'Journal Requirements' section below.* A marked-up copy of your manuscript that highlights changes made to the original version. You should upload this as a separate file labeled 'Revised Manuscript with Track Changes '.* An unmarked version of your revised paper without tracked changes. You should upload this as a separate file labeled 'Manuscript '. If you would like to make changes to your financial disclosure, competing interests statement, or data availability statement, please make these updates within the submission form at the time of resubmission. Guidelines for resubmitting your figure files are available below the reviewer comments at the end of this letter. We look forward to receiving your revised manuscript. Kind regards, Jingjing Yang, Ph.D.Academic EditorPLOS Genetics Xiaofeng ZhuSection EditorPLOS Genetics

Aimée Dudley

Editor-in-Chief

PLOS Genetics

Anne Goriely

Editor-in-Chief

PLOS Genetics

**Journal Requirements:** **Additional Editor Comments (if provided):** Please address all comments from three reviewers.**Reviewers' comments:** Reviewer's Responses to Questions

**Comments to the Authors:**

Reviewer #1: Geneformer is a transformer-based deep learning model designed for single-cell RNA-seq analysis, focusing on learning gene-gene interactions to enable tasks like cell-type classification and gene perturbation prediction. The paper developed a mouse-specific GeneFormer by pretraining Geneformer on a constructed mouse dataset. The architecture and strategy are the same as those that have been used in GeneFormer.

It seems nowadays there’s a trend in developing foundation models in scRNA-seq data, such as Geneformer and scGPT. And the authors’ endeavor to develop a mouse single-cell foundation model is valuable.

I have the following two major comments and a few minor ones.

1. The authors constructed a large-scale dataset of mouse single-cell RNA-seq data i.e., mouse-Genecorpus-20M, derived from healthy mice. Would it be good to include diseased mouse data in the pre-training? From Fig 2, the training data seem unbalanced in terms of ST platform and organ, etc. What would be the author’s comment on this?

2. The authors demonstrated the advantage of mouse-Geneformer on cell type classification and perturbation experiments. I think to make this a potentially widely used foundation model in mice, the author may demonstrate its potential and advantage of it more comprehensively. For example, for mouse single-cell analysis, similar to GeneFormer, it may be helpful to include more tasks.

Minor comments:

When evaluating the cell type classification, would it be suitable to include scGPT?

Fig. 2, maybe a pie chart?

Fig. 4, maybe adjust the size of the dots for better visualization?

Reviewer #2: In this manuscript, the authors collected 20 M single-cell RNA sequencing (scRNA-seq) data from mice and trained a model called mouse-Geneformer, based on the existing Geneformer architecture. The rationale for training this model stems from the widespread use of mice as model organisms in biomedical research. Additionally, the manuscript explores the potential of cross-species applications, demonstrating the promise of mouse-Geneformer for cross-species analysis. Interestingly, it achieves cell type classification accuracies comparable to those of the original human Geneformer model. Their results suggest that mouse-Geneformer can be a valuable tool not only for mouse biology but also for cross-species transcriptomic analysis, potentially benefiting research on non-model organisms and human data inaccessible due to ethical constraints.

With merits being said, despite the large scale of the dataset, the study does not introduce any novel models or algorithms from a methodological perspective, but instead simply replaces the pretraining dataset of the original Geneformer model. This may weaken the manuscript's overall innovation. Furthermore, the evaluation of model performance and cross-species analysis is not sufficiently robust, failing to effectively demonstrate the significance of mouse-Geneformer and its advantages over existing methods. Specific comments are given as follows.

Major

1. Comparison with the original human Geneformer: Has the authors attempted to analyze mouse data using the original Geneformer by fine-tuning with mouse data? If so, did fine-tuning or direct use of the original Geneformer produce results similar to those of mouse-Geneformer in downstream mouse tasks? If comparable results can be achieved, the necessity of developing a mouse-specific version (mouse-Geneformer) might need further justification. Even if mouse-Geneformer outperforms the original Geneformer in certain aspects, the authors should provide a detailed performance comparison between the two models to substantiate the advantages of developing a mouse-specific version.

2. Limitations of cross-species homology-based gene conversion: While gene orthology conversion strategies can be useful in cross-species studies, gene homology alone cannot fully account for biological functional differences between species. Even when genes are homologous, their regulatory mechanisms, spatiotemporal expression patterns, and involvement in biological pathways can vary significantly across species. These differences may contribute to the model's inconsistent performance in cross-species applications (e.g., as noted in the differing COVID-19 results). The authors should discuss which types of analyses this orthology conversion strategy is best suited for. Is it applicable to all genes, particularly those with divergent biological functions across species? What is the practical applicability of this strategy in real biological research? Without deeper discussion of these questions, the reliability of cross-species application of the model may be called into question.

3. Cross-species experiment clarity. While the ortholog-based gene conversion method showed near-perfect accuracy (99.97%) for human cell type classification, this result is unexpectedly high. It would be insightful for the authors to compare this method with other baseline models, such as logistic regression, scVI, or foundation models like scBERT and scGPT, to confirm if the high accuracy is genuinely due to the ortholog conversion or if the task itself is particularly easy.

4. Inconsistent datasets in similar tasks: In the sections 'Cell type classification using the mouse-Geneformer and other methods' and 'Cell type classification using mouse-Geneformer with and without prior learning', the authors use different datasets but do not explain why different datasets are used for the same task type. Specifically, in Table 2, the mouse-Geneformer achieves an accuracy greater than 0.9, whereas in Table 3, the accuracy on two different datasets hovers around 0.7. How do other methods perform on these two datasets in comparison? The lack of consistent dataset usage for similar tasks raises questions about the validity of comparisons across experiments.

5. Limited model comparisons and evaluation metrics: In the 'Cell type classification using the mouse-Geneformer and other methods' task, the authors compare mouse-Geneformer with only two other methods, scDeepSort and scVAE, which seems limited given the broad scope of the field. Additionally, accuracy is the sole evaluation metric presented, which may not sufficiently capture model performance, especially in the presence of class imbalances. Accuracy alone may overlook how the model performs on minority cell types, which are often critical in scRNA-seq analyses. Furthermore, since mouse-Geneformer was fine-tuned on the dataset, what are the results in a zero-shot setting?

Minor:

1. Add more visualizations to show the results. It’s recommended to include additional figures in the paper to better demonstrate key findings, especially for classification and perturbation results.

Reviewer #3: Major concerns.

1. scGPT is an alternative to geneformer. could the authors give some brief discussions on the difference between scGPT and geneformer, and why to build their model based on geneformer instead of scGPT?

2. How are batch effect (e.g., platform) being handled in both training and inference stages?

3. The genetic difference and transcriptional difference of different mouse strains can be huge. How are they handeled in the model.

4. for "Cell type classification using the mouse-Geneformer and other methods", "The mouse data used for comparison comprised single-cell data from nine organs of mice: tongue [25], thymus [25], mammary gland [25], large intestine [25], limb muscle [25], spleen [25], heart [25], brain [25], and kidney [25]." Are these data part of the 20M pre-training data? If not, is there a way to evaluate how similar these data to the training data? the data leaking is always a concern for deep learning. Similar question of data similarity also applies to the "Cell type classification using mouse-Geneformer with and without prior learning".

5. "The mouse-Geneformer was then fine-tuned for disease classification tasks, ensuring that the disease classification accuracy of the test data exceeds 90%.". What kind of disease classification task? the disease should be labeled at the level of mice? but the classification is done on the level of cells?

6. "When deleting a gene, the gene was removed from the dataset, and the rank values of other genes were increased. Conversely, when activating a gene, the rank value of that gene was increased, and the rank values of other genes were decreased. "

First, these statements are confusing. "When deleting a gene, the gene was removed from the dataset". Is it really removed or just set its expression to 0? "the rank values of other genes were increased", should it be "the rank values of the genes with lower ranks were increased"? For the part "when activating a gene, the rank value of that gene was increased, and the rank values of other genes were decreased." How much does the rank value of that gene was increased? again, the rank values of subset of genes rather than all the genes were decreased?

Second are these the procedure to simulate perturbed data? then it is very unrealistic to assume only one gene is perturbed without any effect on other genes?

There are also some minor issues.

1. "3) Cells with fewer than seven detected genes per cell were removed. 4) Cells with more than 20,000 total gene expression levels were excluded. "

Is this too extreme to have only 8-100 genes expressed per cell? "Cells with more than 20,000 total gene expression levels were excluded" means more than 20,000 detected genes?

2. caption of table 1 "Eval 2 is its using" -> "Eval 2 uses"

3. "For cell type classification with scDeepSort, a pretrained deep learning model utilizing weighted Graph Neural Networks (GNNs) for single-cell RNA-seq analysis [29], we employed scDeepSort, which consists of a three-layer graph neural..." this sentence need to be rewritten. There are also other places in the paper where the wording or organization of sentence need some edits.

**Have all data underlying the figures and results presented in the manuscript been provided?**

Reviewer #1: None

Reviewer #2: Yes

Reviewer #3: Yes

PLOS authors have the option to publish the peer review history of their article (what does this mean? ). If published, this will include your full peer review and any attached files.

**Do you want your identity to be public for this peer review?** For information about this choice, including consent withdrawal, please see our Privacy Policy .

Reviewer #1: No

Reviewer #2: No

Reviewer #3: No

---

## [Decision Letter · Decision Letter 1]

6 Feb 2025

PGENETICS-D-24-01035R1

Mouse-Geneformer: A Deep Learning Model for Mouse Single-Cell Transcriptome and Its Cross-Species Utility

PLOS Genetics

Dear Dr. Shigenobu,

Thank you for submitting your manuscript to PLOS Genetics. You can see reviewer 1 and 2 are satisfied with the revision and  reviewer 3 still have some minor concerns. Therefore, we invite you to submit a revised version of the manuscript that addresses the points raised reviewer 3.

Please submit your revised manuscript within 30 days Mar 08 2025 11:59PM. If you will need more time than this to complete your revisions, please reply to this message or contact the journal office at plosgenetics@plos.org. Please include the following items when submitting your revised manuscript:

We look forward to receiving your revised manuscript.

Kind regards,

Jingjing Yang, Ph.D.

Academic Editor

PLOS Genetics

Xiaofeng Zhu

Section Editor

PLOS Genetics

Aimée Dudley

Editor-in-Chief

PLOS Genetics

Anne Goriely

Editor-in-Chief

PLOS Genetics

**Journal Requirements:**

1) We note that the figures are duplicated on your submission. Please remove any unnecessary files. 

2) Figure files in .tif format are currently uploaded as file type “Other”, which is not viewable by the reviewers. Please change the file type(s) to 'Figure."

**Reviewers' comments:**

Reviewer's Responses to Questions

Reviewer #1: Thank you for addressing/discussing my comments.

Reviewer #2: All comments have been addressed.

Reviewer #3: 1. For the question on the genetic difference and transcriptional difference of different mouse strains. The authors answered

"We found that the majority of the data were derived from C57BL/6 mice".

"It is also worth noting that the original human Geneformer was trained on scRNA-seq data from a much broader range of human individuals with significant genetic and transcriptomic diversity. Despite this, the Geneformer architecture demonstrated robust and reliable performance across a wide variety of downstream tasks. Based on this precedent, we expect that the mouse- Geneformer, which is built using a more genetically homogeneous dataset, would similarly exhibit robustness."

The genetic difference between different mouse strains is much larger than the genetic difference between human races. They are comparable to the genetic difference between human and chimpanzees. Therefore, "majority of the data were derived from C57BL/6 mice" is diffrent from all the data were derived from one C57BL/6 mice, and it may not means the model was built using a more genetically homogeneous dataset. I understand it is a lot to ask to trace down the genetic origin of each dataset. At least the authors should acknowledge that this is a potential concern, and the users should keep the mouse strain in mind when using this model.

2. (3-6B)

Second are these the procedure to simulate perturbed data? then it is very unrealistic to assume only one gene is perturbed without any effect on other genes?

"Ans) In the in silico perturbation experiments, we aim to mimic gene knockout experiments in the laboratory, where a specific target gene is disrupted. Typically, such experiments involve perturbation of a single gene, and our approach reflects this. In our model, removing a single gene from the gene list using the Rank Value Encoding method corresponds to simulating this gene knockout scenario."

"As the reviewer rightly pointed out, deleting a single gene is expected to affect the expression of many other genes due to downstream regulatory interactions. These secondary effects are precisely what the Geneformer models are designed to predict, capturing how the perturbation of one gene influences broader gene expression networks. Therefore, while we simulate the perturbation of a single gene, the model inherently accounts for the cascade of transcriptional changes that follow."

Maybe I miss something here. The question I asked was how the author simulated the expression change of other genes? If the expression change of other genes were not simulated, but predicted by Geneformer, how the authors evaluated such prediction?

**Have all data underlying the figures and results presented in the manuscript been provided?**

Reviewer #1: None

Reviewer #2: Yes

Reviewer #3: None

PLOS authors have the option to publish the peer review history of their article (what does this mean? ). If published, this will include your full peer review and any attached files.

**Do you want your identity to be public for this peer review?** For information about this choice, including consent withdrawal, please see our Privacy Policy .

Reviewer #1: No

Reviewer #2: No

Reviewer #3: No

**Figure resubmission:**
---

## [Decision Letter · Decision Letter 2]

17 Feb 2025

Dear Dr Shigenobu,

We are pleased to inform you that your manuscript entitled "Mouse-Geneformer: A Deep Learning Model for Mouse Single-Cell Transcriptome and Its Cross-Species Utility" has been editorially accepted for publication in PLOS Genetics. Congratulations!

Yours sincerely,

Jingjing Yang, Ph.D.

Academic Editor

PLOS Genetics

Xiaofeng Zhu

Section Editor

PLOS Genetics

Aimée Dudley

Editor-in-Chief

PLOS Genetics

Anne Goriely

Editor-in-Chief

PLOS Genetics

Comments from the reviewers (if applicable):

All reviewers are now satisfied with the revision.

Reviewer's Responses to Questions

**Comments to the Authors:**

Reviewer #3: the authors have addressed my questions.

**Have all data underlying the figures and results presented in the manuscript been provided?**

Reviewer #3: None

PLOS authors have the option to publish the peer review history of their article (what does this mean? ). If published, this will include your full peer review and any attached files.

**Do you want your identity to be public for this peer review?** For information about this choice, including consent withdrawal, please see our Privacy Policy .

Reviewer #3: **Yes: ** Wei Sun

**Data Deposition**

http://datadryad.org/submit?journalID=pgenetics&manu=PGENETICS-D-24-01035R2

**Press Queries**

---

## [Editor Report · Acceptance letter]

PGENETICS-D-24-01035R2

Mouse-Geneformer: A Deep Learning Model for Mouse Single-Cell Transcriptome and Its Cross-Species Utility

Dear Dr Shigenobu,

We are pleased to inform you that your manuscript entitled "Mouse-Geneformer: A Deep Learning Model for Mouse Single-Cell Transcriptome and Its Cross-Species Utility" has been formally accepted for publication in PLOS Genetics! Your manuscript is now with our production department and you will be notified of the publication date in due course.

With kind regards,

Anita Estes

PLOS Genetics

On behalf of:
